# Peptidomics Unveils Distinct Acetylation Patterns of Histone and Annexin A1 in Differentiated Thyroid Cancer

**DOI:** 10.3390/ijms25010376

**Published:** 2023-12-27

**Authors:** Margarida Coelho, João Capela, Vera M. Mendes, João Pacheco, Margarida Sá Fernandes, Isabel Amendoeira, John G. Jones, Luís Raposo, Bruno Manadas

**Affiliations:** 1CNC—Center for Neurosciences and Cell Biology, University of Coimbra, 3004-504 Coimbra, Portugal; mmcoelho@cnc.uc.pt (M.C.);; 2CIBB—Centre for Innovative Biomedicine and Biotechnology, University of Coimbra, 3004-504 Coimbra, Portugal; 3III Institute for Interdisciplinary Research (IIIUC), University of Coimbra, 3030-789 Coimbra, Portugal; 4Department of Chemistry, Faculty of Sciences and Technology, University of Coimbra, 3004-535 Coimbra, Portugal; 5Centro Hospitalar Universitário São João, 4200-319 Porto, Portugal; 6I3S, Instituto de Investigação e Inovação em Saúde, 4200-135 Porto, Portugal; 7Ipatimup, Instituto de Patologia e Imunologia Molecular da Universidade do Porto, 4200-465 Porto, Portugal; 8Portuguese Society of Endocrinology, Diabetes and Metabolism, 1600-892 Lisbon, Portugal; 9EPIUnit, Institute of Public Health, University of Porto, 4050-600 Porto, Portugal

**Keywords:** peptidomics, proteomics, tissue, LC-MS/MS, biomarker, thyroid cancer

## Abstract

Thyroid cancer is a common malignancy of the endocrine system. Nodules are routinely evaluated for malignancy risk by fine needle aspiration biopsy (FNAB), and in cases such as follicular lesions, differential diagnosis between benign and malignant nodules is highly uncertain. Therefore, the discovery of new biomarkers for this disease could be helpful in improving diagnostic accuracy. Thyroid nodule biopsies were subjected to a precipitation step with both the insoluble and supernatant fractions subjected to proteome and peptidome profiling. Proteomic analysis identified annexin A1 as a potential biomarker of thyroid cancer malignancy, with its levels increased in malignant samples. Also upregulated were the acetylated peptides of annexin A1, revealed by the peptidome analysis of the supernatant fraction. In addition, supernatant peptidomic analysis revealed a number of acetylated histone peptides that were significantly elevated in the malignant group, suggesting higher gene transcription activity in malignant tissue. Two of these peptides were found to be robust malignancy predictors, with an area under the receiver operating a characteristic curve (ROC AUC) above 0.95. Thus, this combination of proteomics and peptidomics analyses improved the detection of malignant lesions and also provided new evidence linking thyroid cancer development to heightened transcription activity. This study demonstrates the importance of peptidomic profiling in complementing traditional proteomics approaches.

## 1. Introduction

Thyroid nodules are extremely common, but the majority are benign, with a malignancy rate of only 1% [1]. Most thyroid malignancies are differentiated thyroid carcinomas [2] from follicular cells, either papillary (80–85% of cases) or follicular carcinomas (10–15% of cases). The standard diagnostic procedure is fine needle aspiration biopsy (FNAB) followed by a cytological analysis with diagnosis being made according to the categories of the Bethesda System for Reporting Thyroid Cytopathology. These six diagnostic categories recommend management strategies (e.g., molecular testing, repeat FNAB vs. surgery), and their implied risks of malignancy. The stated goals of this protocol are to inform conservative management of thyroid nodules unlikely to cause harm, but at the same time evaluate the probabilities of malignancy at each stage of analysis [3]. Considering that the initial FNAB cytological analysis is currently inefficient at resolving malignant from benign nodules—mainly in follicular patterned lesions—the development of complementary diagnostic methods that could more effectively discriminate between malignant and benign lesions would be useful.

Although thyroid cancer has been studied by proteomics [4,5], peptidomic approaches remain relatively unexplored, particularly in tissue [6]. Peptides can either originate from the breakdown of a protein or be intentionally expressed to play independent regulatory roles [7]. Peptidomics is the screening of such endogenous peptides, which among other things can potentially provide additional molecular insights into disease etiologies.

This work analysed thyroid nodules with both proteomics and peptidomics approaches using liquid chromatography coupled with tandem mass spectrometry (LC-MS/MS). Alongside the conventional approach of identifying proteins whose levels were different between malignant and benign tissues, this approach also discovered naturally occurring peptides that, in addition to informing gene transcription processes in malignant cells, are also strong candidates for standalone biomarkers of malignancy.

## 2. Results

Thyroid samples were subjected to a protein precipitation procedure, resulting in a sediment composed mainly of proteins and a supernatant composed mainly of peptides and other small molecules (Figure 1a). A total of 2423 proteins were quantified in the proteomics screening, while 149 peptides belonging to 59 proteins were quantified from the supernatant and analysed in a peptidomic screening. The separation of malignant from benign samples observed in the PCAs of both proteomics and peptidomics screening reveals potential differences to be further explored in both datasets (Figure 1b,c). The top 20 statistically different proteins and peptides relative to the *p*-value are depicted in Appendix A and in a heatmap in Appendix A.

Of the 1532 proteins that were considered statistically different, annexin A1 was increased in the malignant group with a fold change of 5.4 and a *p*-value below 0.0001 and presented the highest area under the curve (AUC) in the ROC analysis with 0.992 (95% confidence interval (CI) 0.967–1), demonstrating the potential of this protein to distinguish benign from malignant thyroid tissues [8]. When searching if peptides of this protein were found in the supernatant fraction, three non-tryptic peptides were detected, all statistically different with similar trends as the protein found in the pellet (Figure 2). However, none of these peptides found in the supernatant under non-denaturing conditions was identified in the 14 peptides that were used to quantify this protein in the pellet, nor were they identified in any proteomics sample. Interestingly, these three peptides have a common amino acid sequence and an acetylation.

In the peptidomics screening of the supernatant fraction, 75 peptides were considered statistically different. A total of 11 peptides from three histones appeared elevated in the malignant group compared to the benign (Figure 3). With the exception of one peptide, the other ten were acetylated. After individual ROC analysis, two peptides presented an AUC greater than 0.95: peptide [1Ac]-SETAPLAPTIPAPAEKTPVKK from histone H1.3 with an AUC of 0.967 (95% CI 0.901–1) and peptide SETAPAAPAAPAPAEKTPVK from histone H1.4 with an AUC of 0.956 (95% CI 0.889–1). Proteomics screenings depend highly on protein dynamic range. In our case, these histone peptides were not quantified in the proteomics analysis of the pellet fraction, although the histone H1.4 peptide has been identified in a few proteomics samples, emphasizing the importance of the peptidomics analysis of the supernatant.

Both datasets were also compared to find which peptide sequences presented contradictory behaviours. Although Poly(rC)-binding protein 1 was not found statistically different in the proteomics analysis, the peptide with the sequence FAGIDSSSPEVKG of this protein was upregulated in the supernatant fraction (Figure 4b). In contrast, the peptide QQSHFAMMHGGTGFAGIDSSSPEVK of this protein, with part of the same amino acid sequence (underlined), was found downregulated in the protein fraction (Figure 4a). In addition, no correlation was found between this peptide and the protein (Figure 4c). Considering that this analysis is based on naturally occurring peptides, the cleavage of the peptide before the last phenylalanine could be performed by pepsin according to the Expasy Peptide Cutter tool and by cathepsin B according to TopFIND and MEROPS. The shift in regulation from one peptide to the other suggests that this enzyme could be more active in the malignant participants than in the benign.

## 3. Discussion

Proteomics screenings have been previously applied in the study of thyroid cancer tissues [4,5,9,10], but to our knowledge there are no reports of peptidomics screening. Single peptides can be produced by the breakdown and turnover of regular proteins or by cleavage of a pro-hormone sequence generating a bioactive entity on its own [6]. Therefore, alterations to these peptides can also inform about disease processes such as thyroid cancer. In this work, proteomics and peptidomics approaches were applied to look for differences between benign and malignant thyroid nodules and explore the relationships between proteins and native peptides. Differences between the benign and malignant groups were found in both the proteome and the peptide screenings of these samples (Figure 1, Appendix A).

A specific upregulation of annexin A1 in carcinomas of follicular cell origin has been reported previously [11,12,13] and suggests that this protein could be a diagnostic and prognostic biomarker for thyroid cancer. This calcium and phospholipid-binding protein participates in inflammatory processes, cell proliferation modulation, cell death regulation and tumour formation and development. The proteomics analysis in this work confirms an upregulation of annexin A1 in the malignancy group, but the peptidomics analysis of the same samples revealed corresponding alterations in stand-alone peptides that were not detected in the proteomics screening (Figure 2).

The peptides of three histones were found to be statistically different in malignancy (Figure 3). Moreover, most of these peptides were acetylated, and it is known that this post-translational modification has a role in regulating inflammation, one of the major hallmarks of cancer [14]. Post-translational modifications in histones regulate accessibility to DNA and consequently gene transcription. Acetyl groups neutralize the histones’ positive charges, opening the chromatin structure and allowing transcription machinery to bind to DNA. In this case, a higher histone acetylation expression, such as in most histone peptides identified in this work, suggests higher gene transcription in malignant thyroid nodules. Moreover, this modification has been previously found in the histones of cancer cells, including thyroid cancer [15,16]. Based on these results, it would also be interesting to assess the activity of histone acetyltransferases, as it has been implicated in thyroid cancer [17,18]. Furthermore, histone cleavage/regulation by enzymatic activity as a mechanism to increase gene transcription deserves further study.

The study of peptides can be informative based on their individual roles in disease, and analysis of their amino acid sequence can also inform about differential enzyme activity. Part of a peptide of poly(rC)-binding protein 1 detected in the pellet fraction was detected in the supernatant but with an opposite profile (Figure 4). Considering which enzyme(s) could have been responsible for this specific cut, pepsin is at first sight an unlikely candidate since it is expressed in the stomach. Nonetheless, considering that thyroid hormones can influence pepsin secretion, pepsin activity could be modified in thyroid cancer [19]. Pepsin has also been associated with laryngopharyngeal cancer [20,21], not only by damaging epithelial cells but also by disruption of signalling pathways [22]. Such mechanisms might also be active in thyroid malignancy. However, cathepsin B is the most likely candidate since this proteinase is frequently found to be increased in cancer and is thought to contribute to invasive and metastatic properties of cancer [23], and it has also been previously associated with thyroid cancer [24,25]. The results of this work suggest a higher enzymatic activity in malignant patients, which requires further validation. On the other hand, poly(rC)-binding protein 1 is an RNA-binding protein that can function as a tumour suppressor, having been found to be downregulated in many cancer types [26]. Therefore, increased degradation of this protein could dysregulate gene transcription to promote tumorigenesis, raising the question of whether inhibition of the protease that cleaves poly(rC)-binding protein can act as a tumour suppressor.

Peptides are more susceptible to degradation in biological matrices compared to proteins. In that sense, one limitation of this work is related to the difficulties in performing rapid freezing or heat inactivation to decrease enzyme activity. However, samples were all processed using the same protocol and kept at low temperatures until storage at −80 °C, thereby limiting enzymatic activity. Also, protease inhibitors were used to prevent further protein degradation when first thawed.

## 4. Materials and Methods

### 4.1. Patients

This study was approved by the Ethics Committee of the Centro Hospitalar Universitário São João/Faculdade de Medicina da Universidade do Porto (approval ID 125/18). Informed consent was obtained from all participants. The cohort of thyroid tissue lesions consisted of 71 nodules from 43 patients. The benign group consisted of follicular adenoma and follicular nodular disease, while the malignant group was comprised of differentiated carcinomas of the follicular cells (papillary carcinoma and follicular carcinoma). In some cases, more than one nodule per individual was studied. Groups were gender- and age-matched (Table 1). Tissue samples were obtained during surgical resection and immediately stored at −80 °C until analysis. The final diagnosis was obtained after a postoperative histopathological examination of the same lesion (Appendix A).

### 4.2. Peptidomics

Samples were initially analysed by non-destructive high-resolution magic angle spinning (HR-MAS) ^1^H nuclear magnetic resonance (NMR) for metabolomics screening (not shown in this manuscript), in combination with proteomics. Nodules were recovered from the rotor and stored in a 100 μL 0.5 M triethylammonium bicarbonate (TEAB) solution with protease inhibitors (cOmplete™, ethylenediamine tetraacetic acid (EDTA)-free Protease Inhibitor Cocktail, Roche). Samples not analysed by HR-MAS NMR were also added to the same solution. Tissue samples were homogenised with the Dispersing-aggregates POLYTRON^®^ PT1200 E with a 3 mm tip (Kinematica AG, Malters, Switzerland). The homogenised mixture was centrifuged at 5000× *g* for 5 min at 4 °C. The supernatant was harvested, with 5 μL being used for total protein content assessment with the Pierce™ 660 nm Protein Assay Reagent (ThermoFisher™, Waltham, MA, USA), according to the manufacturer’s instructions. A volume corresponding to approximately 100 μg was harvested from each sample to continue sample processing. Moreover, pools of benign and malignant lesions were created using 5 μL from selected samples. To each individual and pooled sample, 2 μg of the recombinant protein green fluorescent protein and maltose-binding periplasmic protein (MBP-GFP) were added [27]. Protein precipitation was performed using 400 μL of cold methanol. Samples were incubated overnight at −80 °C and then centrifuged at 20,000× *g* for 20 min at 4 °C. The supernatant was harvested for peptidomics analysis, while the pellet was used in the proteomics analysis. After evaporation of the supernatant using the Concentrator Plus/Vacufuge^®^ (Eppendorf, Hamburg, Germany), the supernatant fraction was stored at −80 °C until analysis, where it was resolubilized in 35 μL of 2% acetonitrile and 0.1% formic acid containing 1 μM of the standard sulfamethazine-^2^H_4_ (Toronto Research Chemicals, North York, ON, Canada) aided by sonication with a Vibra cell 75041 cup horn (Bioblock Scientific, Illkirch, France) at 20% amplitude every 1 s pulse for 2 min. After centrifugation at 14,100× *g* for 5 min at room temperature, samples were transferred to vials for LC-MS analysis. After individual sample resuspension, five additional pools were created from 5 μL of a selection of prepared samples for identification purposes. Samples were analysed on a NanoLC™ 425 System (Eksigent^®^, Framingham, MA, USA) coupled to a TripleTOF™ 6600 System (Sciex^®^, Framingham, MA, USA) using data-dependent acquisition (DDA) on pooled samples for peptide identification and data-independent acquisition (DIA/SWATH-MS) acquisition of each individual sample for peptide quantification. Detailed procedures of data acquisition are described in Appendix B (Materials and Methods A1). Peptide identification and library generation were performed with ProteinPilot™ 5.0 (Sciex^®^), while data processing for quantification was performed using the SWATH™ processing plug-in for PeakView™ 2.2 (ABSciex^®^). Detailed procedures of data processing are described in Appendix B (Materials and Methods A2). The peptidomics data have been deposited to the EMBL-EBI MetaboLights database [28] with the identifier MTBLS5206.

### 4.3. Proteomics

The pellet fraction, containing the proteins, was resuspended in 30 μL of 2× Laemmli Sample Buffer by sonication with a Vibra cell 75041 cup horn (Bioblock Scientific, Illkirch, France). Samples were incubated at 95 °C for 5 min in a Thermomixer comfort (Eppendorf, Hamburg, Germany) and 2 μL of 40% acrylamide (Bio-Rad Laboratories, Lda., Hercules, CA, USA) were added as an alkylating agent. Individual and pooled samples were loaded into an SDS-PAGE 4–20% Mini-PROTEAN^®^ TGX™ precast gel and resolved at 110 V [29]. Gel staining was performed as previously described [30]. Each lane was divided into fractions, and each was divided into smaller pieces with the help of a scalpel and added to a 96 multi-well plate containing ddH_2_O. After destaining the gel pieces with a 50 mM ammonium bicarbonate solution and 30% acetonitrile, in-gel digestion and peptide extraction were performed as previously described [30]. Peptides were evaporated in the Concentrator Plus/Vacufuge^®^ (Eppendorf, Hamburg, Germany) and resolubilized in 30 μL of 2% acetonitrile and 0.1% formic acid aided by sonication with a Vibra cell 75,041 cup horn (Bioblock Scientific, Illkirch, France) at 20% amplitude every 1 s pulse for 2 min. After centrifugation at 14,100× *g* for 5 min at room temperature, samples were transferred to vials for LC-MS analysis. Samples were analysed on a NanoLC™ 425 System (Eksigent^®^, Framingham, MA, USA) coupled to a TripleTOF™ 6600 System (Sciex^®^) using DDA for each fraction of the pooled samples for protein identification and DIA/SWATH-MS acquisition of each individual sample for protein quantification. Detailed procedures of data acquisition are described in Appendix B (Materials and Methods A3). Peptide identification and library generation were performed with ProteinPilot™ 5.0 (Sciex^®^), while data processing for quantification was performed using the SWATH™ processing plug-in for PeakView™ 2.2 (ABSciex^®^). Detailed procedures of data processing are described in Appendix B (Materials and Methods A4). The proteomics data have been deposited to the ProteomeXchange Consortium via the PRIDE [31] partner repository with the dataset identifier PXD035583.

### 4.4. Data Analysis

Multivariate analysis was performed in Metaboanalyst 5.0 (https://www.metaboanalyst.ca) [32]. Log-transformation and Pareto scale were performed for principal component analysis (PCA) and partial least squares discriminant analysis (PLS-DA). All ellipses in the scores plots for both PCA and PLS models, were drawn at the 95% confidence level. Receiver operating characteristic (ROC) analysis was also performed on this platform, with the same normalization parameters. Multivariate ROC curves were based on a linear support vector machine (SVM) classification model and SVM built-in feature ranking method. Heatmaps with hierarchical clustering used a Euclidean distance measure and Ward clustering method.

Correlations and linear regressions were performed on GraphPad Prism 6.0 (GraphPad Software Inc., San Diego, CA, USA). Due to the small number of samples in each group and the lack of normal distribution of the populations, a Mann–Whitney test was applied to test differences between groups using IBM SPSS Statistics 23 (IBM^®^, Armonk, NY, USA).

Prediction of potential protease or chemical cleavage sites in a protein sequence was performed on Expasy Peptide Cutter web tool [33], TopFIND [34] and MEROPS [35].

## 5. Conclusions

This work demonstrates how different analytical approaches on the same samples can provide additional data, in this case uncovering small peptides that would otherwise have gone unnoticed with proteomics alone. In addition, the comparison of peptidomics and proteomics results from thyroid nodules revealed different results at the peptide level that can also be explored in the future. The peptidomics approach can also indicate potential enzymatic activity, which requires further validation. Regarding biomarker potential, the peptides themselves could be searched in the aspirates of FNAB and/or in blood samples as tissue or as peripheral biomarkers using targeted approaches.

## Figures and Tables

**Figure 1 ijms-25-00376-f001:**
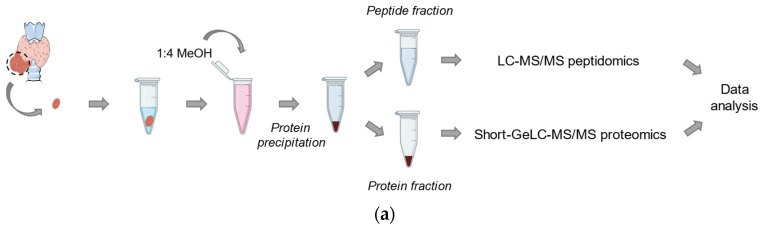
Workflow and principal component analysis (PCA) of benign and malignant samples in both the proteomics screening from the pellet fraction and the peptidomics screening of the supernatant: (**a**) schematic of sample preparation and analysis of the protein and peptide fractions; PCA scores plot computed using multivariate analysis of (**b**) proteomics data and (**c**) peptidomics data for benign (grey) and malignant (black) thyroid lesions.

**Figure 2 ijms-25-00376-f002:**
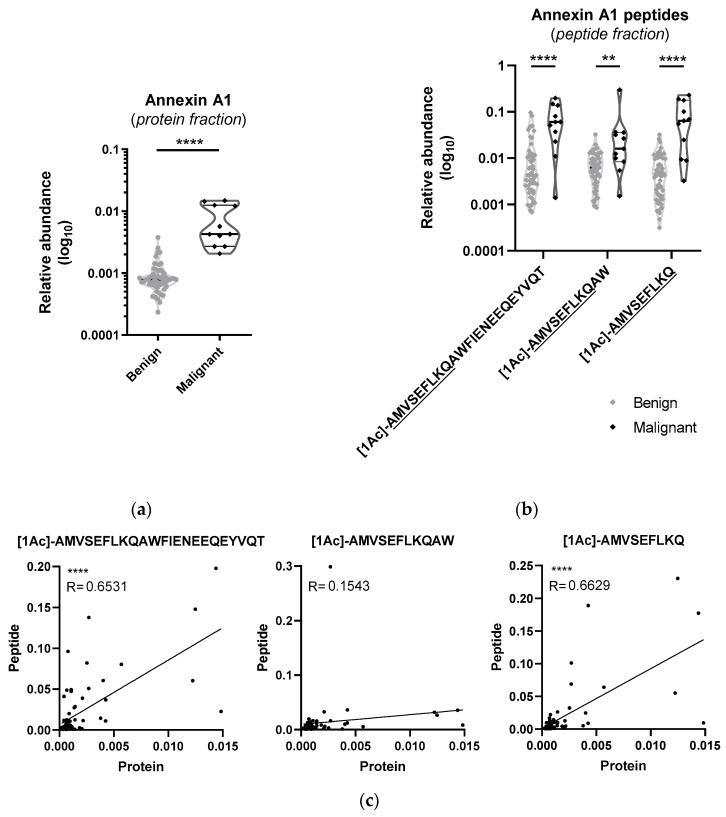
Annexin A1 peptides found in the supernatant were increased, similarly to the protein quantification obtained in the proteomics screening: violin plots with individual sample representation of (**a**) annexin A1 protein from the proteomics analysis and (**b**) peptides of annexin A1 from the peptidomics analysis of the peptide fraction of the same tissue samples. Peptide sequences are represented by amino acid one letter code and acetylation at the N-termini is represented by [1Ac]. Benign samples are represented in grey and malignant ones in black. ** *p* ≤ 0.01, **** *p* ≤ 0.0001 (Mann–Whitney test). (**c**) Correlations between annexin A1 protein from the proteomics analysis and each peptide from the peptidomics analysis. **** *p* ≤ 0.0001 (Pearson correlation).

**Figure 3 ijms-25-00376-f003:**
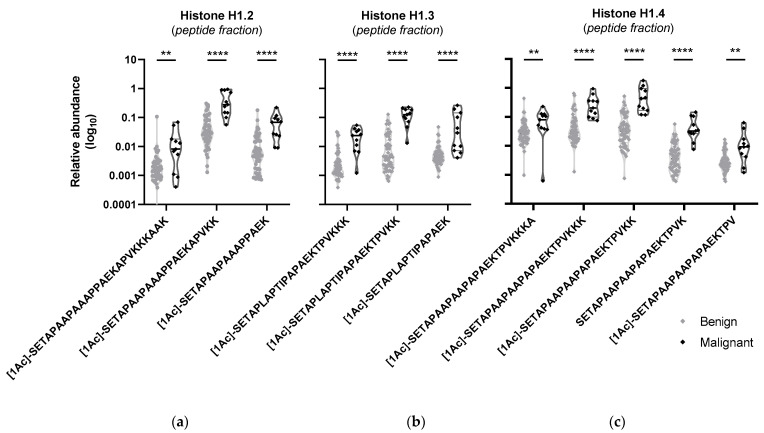
Peptides and acetylated peptides from histones were found increased in malignant samples: violin plots with individual sample representation of peptides of histones (**a**) H1.2, (**b**) H1.3 and (**c**) H1.4 from the peptidomics analysis of the peptide fraction of tissue samples of benign versus malignant groups. Peptide sequences are represented by amino acid one letter code and acetylation at the N-termini is represented by [1Ac]. Benign samples are represented in grey and malignant ones in black. ** *p* ≤ 0.01, **** *p* ≤ 0.0001 (Mann–Whitney test).

**Figure 4 ijms-25-00376-f004:**
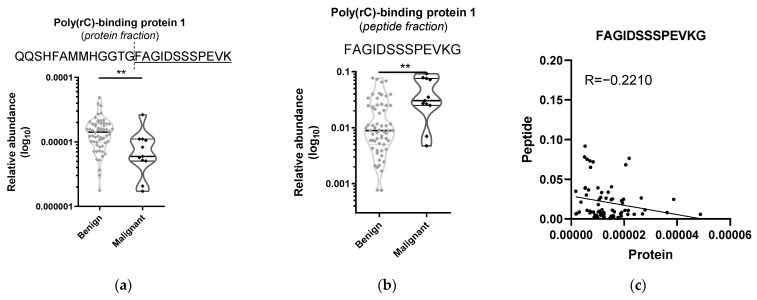
Differential abundance in the same amino acid sequence of Poly(rC)-binding protein 1 suggests differences in enzymatic activity: violin plots with individual sample representations of (**a**) Poly(rC)-binding protein 1 peptide from the proteomics analysis and (**b**) part of that same peptide from the peptidomics analysis of the peptide fraction of the same tissue samples, of benign versus malignant groups. Peptide sequences are represented by amino acid one letter code. Benign samples are represented in grey and malignant ones in black. The dashed vertical line in the peptide sequence represents a possible cleavage site. ** *p* ≤ 0.01 (Mann–Whitney test). (**c**) Correlation between poly(rC)-binding protein 1 from the proteomics analysis and peptide FAGIDSSSPEVKG from the peptidomics analysis. Non-significant (Pearson correlation).

**Table 1 ijms-25-00376-t001:** Demographic summary of the sample cohort.

Group	Sex	Female (%)	Age (Years)	BMI (kg/m^2^)	Free T4 (ng/dL)	TSH (um/L)
Benign	Female = 24 Male = 9	72.7	58.4 ± 2.3	28.7 ± 0.9 **	1.11 ± 0.07	0.83 ± 0.12
Malignant	Female = 8 Male = 2	80.0	53.2 ± 7.1	23.7 ± 1.0	1.08 ± 0.04	1.28 ± 0.29

Note: Data represents mean ± standard error of mean. ** *p* ≤ 0.01 (Mann–Whitney test). Abbreviations: body mass index (BMI), thyroid-stimulating hormone (TSH) and Thyroxine (T4).

## Data Availability

The data presented in this study are openly available in the EMBL-EBI MetaboLights database with the identifier MTBLS5206 and the ProteomeXchange Consortium via the PRIDE partner repository with the dataset identifier PXD035583.

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
