# Peer review of "Peptidomics Unveils Distinct Acetylation Patterns of Histone and Annexin A1 in Differentiated Thyroid Cancer"

_ijms, 2023, doi:10.3390/ijms25010376_

Round 1

Reviewer 1 Report

Comments and Suggestions for Authors

The manuscript titled "Peptidomics unveils distinct acetylation patterns of histone and annexin A1 in differentiated thyroid cancer" by Margarida Coelho et al. Below are the concerns that would need to be addressed.

- Annexin A1 is reported to be overexpressed in many different types of human cancers. It is therefore not surprising to see that Annexin A1 was also identified to be upregulated in thyroid cancer. So the authors reported different degree of acetylation levels were observed in Annexin A1 when comparing benign and malignant samples. It would be nice to draw a schematic showing which lysines are acetylated, and the species and length of Annexin A1 peptides detected in the supernatant.

- While I agree MS is a powerful approach to dig out the potential biomarkers from thousands of proteins present in the sample, yet, after the identification of potential biomarkers by MS, it would require further validation by either immunoblot analysis or ELISA in order to support the MS result.

- For the supplementary table S1, it is unclear what are samples A, B and C? Please define them clearly in the footnote. Also, please add more information regarding the gender, age and ethnicity of patients individually.

- It would be nice if there is another bigger number cohort of thyroid cancer patients and testing the accuracy and reliability of the biomarkers or protein/peptide signatures identified by MS (by single-blind manner preferrably).

- At the end, it remains unclear to me why the authors deliberately focus the targets present in both the supernatant and precipitated fractions? Any more intersesting and novel targets that are present only in supernatant/precipitated fractions? Anyhow, I think it would be essential to list a table in the main text showing all the identified targets in the supernatant and precipitated fraction (even the top 20s would be nice).

Comments on the Quality of English Language

Typos and unfriendly mode of English usage can be found.

Reviewer 2 Report

Comments and Suggestions for Authors

The manuscript has a very interesting premise, the problem to determine malignancy is a very important issue to address malignancy in FNAB and a new way to investigate it, through the combination of peptidomics and proteomics.

However, the results seem not really explored in deep and the number of samples is small taking into account that thyroid nodules are “extremely common”. Here are some improvements on the manuscript that I believe will make the manuscript more complete.

1.      Figure 1 is cut.

2.      Perform a heatmap with clusterization +  a dendogram with differentially present peptides and proteins, to differentiate between benign and malignant samples.

3.      Perform a gene ontology analysis with DE proteins.

4.      Compare your results with other proteomics analysis (for example Martínez-Aguilar, J., Clifton-Bligh, R. & Molloy, M. Proteomics of thyroid tumours provides new insights into their molecular composition and changes associated with malignancy. Sci Rep 6, 23660 (2016).

5.      Show the list of statistically different proteins and peptides in a table, with fold change, p-value, corrected p-value.

6.      Justify a little the focus in Annexin A1 and histones, and the sudden appearance of PolyrC-binding-protein 1.

7.      Compare your results with transcriptomics studies for example TCGA or other.

8.      It would be an asset to the paper to make the results of the counts of the proteins and peptide available for external validation. In the same article as a supplementary or in a repository.

Round 2

Reviewer 1 Report

Comments and Suggestions for Authors

Although the authors explained why validation of Annexin A1 was not performed, however, I believe this is a essential data. MS data cannot provide the full length protein size observation. Only through immunoblot analysis can allow one to examine the protein identity and whether the molecular size of the target band. I insist that at least few pairs of normal and patient tissues should be run for immunoblot analysis in order to provide support to the MS data.

Comments on the Quality of English Language

Typos can still be found.
